# Agile THz-range spectral multiplication of frequency combs using a multi-wavelength laser

Shahab Abdollahi [1] ✉, Mathieu Ladouce [1], Pablo Marin-Palomo [1] & Martin Virte [1] ✉

A breakthrough technology, on-chip frequency comb sources offer broadband combs while being compact, energy-efficient, and cost-effective solutions for various applications from lidar to telecommunications. Yet, these sources encounter a fundamental trade-off between controllability and bandwidth: broadband combs, generated in microresonators, lack free-spectral range or spectral envelope control, while combs generated with electro-optic modulators can be carefully tailored but are limited in bandwidth. Here, we overcome this trade-off through agile spectral multiplication of narrowband combs. Exploiting the nonlinear dynamics of a multi-wavelength laser under modulated optical injection, we achieve spectral multiplication at frequency offsets from 26 GHz to 1.3 THz. Moreover, on-chip control allows for nanosecond switching of the frequency offset. Compatible with generic platforms, our approach can be scaled up to cover several THz. When combined with THz photomixers, our system could enable low-cost, compact, and power-efficient THz comb sources, paving the way towards a new generation of THz applications.

Optical frequency combs (OFCs) have been identified as a key building block for many applications ranging from spectroscopy[1,2] and lidar[3] to optical communications[4–6]. The ubiquity and versatility of frequency combs naturally made them a crucial field of research, especially in photonics as tremendous performances can be reached thanks to the intrinsic advantages of light and light technology. Several photonic techniques and platforms have been demonstrated for generating frequency combs covering the visible to the THz spectral range[7–11]. Combined with the progress of photonic integration, highly compact, relatively cheap, energy-efficient combs sources are now accessible[12–14]. There is, however, an inherent trade-off between current OFC sources: large comb bandwidths typically come at the price of no or limited tunability, meaning that a system tends to be optimized for one specific use case. In this sense, having a single device with the ability to tune the comb spacing or control the spectral envelope or even the coherence would be highly desirable so that one

system could match the requirements of various applications. We identify four parameters of OFC sources whose tunability would be a strong plus: 1) The free-spectral range (FSR), i.e. the spectral comb line spacing, e.g. for flexible grid transmission networks[15–17]. 2) The frequency comb centre wavelength, e.g., for spectroscopy applications[18]. 3) The frequency envelope, to direct all the power to the frequency range of interest, e.g., for THz generation[19]. 4) The coherence of the comb lines, e.g., for lidar or ranging applications for which broad linewidth can aid absolute positioning[20]. Different approaches have been considered to adjust the properties of the OFC. One can start from a broad frequency comb and process it to achieve the desired features. The envelope of the comb can relatively easily be tuned with frequency filters, but the FSR is fixed by the initial comb source. For instance, mode-locked lasers and micro-cavities generate broadband combs with excellent properties, but the corresponding FSR is intrinsically fixed[21] and small FSR values, typically below the GHz, represent a

---

[1]Brussels Photonics Team (B-PHOT), Vrije Universiteit Brussel (VUB), Pleinlaan 2, 1050 Brussel, Belgium. ✉e-mail: mohammadshahab.abdollahi@vub.be; martin.virte@vub.be

challenge with these approaches. On top, optical filters also suffer from limited bandwidth, and centre frequency control requires complex setups, hindering photonic integration[22–25]. Even more importantly, the power of filtered frequencies is totally lost. On the other hand, nonlinear approaches to perform more advanced comb processing or conversion tasks are promising and flexible but are typically energy inefficient as they require high power input[26]. Alternatively, electro-optic (EO) comb sources are more straightforward and flexible, but they are facing major limitations in terms of frequency range: reaching mm-wave frequencies is already an issue, thus keeping most of the THz range out of reach for now. To overcome these hurdles, one requires complex systems and techniques such as cascading EO phase-modulator devices together with highly nonlinear fibers[27] or microresonators[28]. Last, recent works showed the potential of optical injection schemes in semiconductor lasers to unlock comb processing capabilities. Injecting an optical frequency comb into a semiconductor laser can lead to comb broadening and changes in the polarization properties of the comb[29,30].

In this work, we demonstrate frequency-agile spectral multiplication of an EO-comb by a semiconductor multi-wavelength laser (MWL). Our novel scheme combines the flexibility of the EO-comb with the large bandwidth of a multi-wavelength laser to obtain a versatile OFC with a flexible spectral envelope over THz-range frequencies. Through optical injection of a narrowband comb, we achieve comb multiplication from tens of GHz up to 1.3 THz, while preserving the RF coherence of the injected comb, i.e. the phase correlation between its comb lines. However, phase correlation between the various sub-combs is not achieved per se. As such, we refer to this process as spectral multiplication rather than comb broadening. We further

demonstrate that the different multiplied combs can be turned on and off on demand with over 30 dB side-mode suppression ratio. The control mechanism is realized by adjusting the phase in a tailored optical feedback cavity monolithically integrated with the laser, allowing switching at the nanosecond time scale. Moreover, we show that phase locking between the multiplied combs can be achieved either by adding an extra tone to the injected signal or carefully adjusting the tone spacing of the injected comb, thus leading to "cascaded" phase locking of neighboring combs. Finally, we report an excellent agreement with numerical simulations based on a multi-mode rate equation model. Besides reproducing the experimental results, our numerical investigations highlight the crucial importance of the coupling between the different wavelengths of the MWL to enable spectral multiplication. This work represents a novel direction towards high-speed versatile on-chip optical processing of optical signals, not only limited to frequency combs, over THz spectral ranges.

## Results

### THz-scalable spectral multiplication of a frequency comb with an on-chip MWL

The concept we propose for on-chip comb spectral multiplication with an integrated multi-wavelength laser (MWL) is depicted in Fig. 1. The inherent emission of the MWL consists of multiple modes ($\lambda_1, \lambda_2, ..., \lambda_n$) which can be flexibly designed to be spectrally separated from a few GHz to several THz; the main limit being the gain bandwidth of the associated active medium, e.g., up to 10 THz around 1550 nm in the InP platform[31]. A control mechanism enables emission at a specific $\lambda_n$ or even at multiple wavelengths, e.g., $\lambda_n$ and $\lambda_m$, simultaneously. To achieve spectral multiplication, a narrow comb, i.e., with a bandwidth

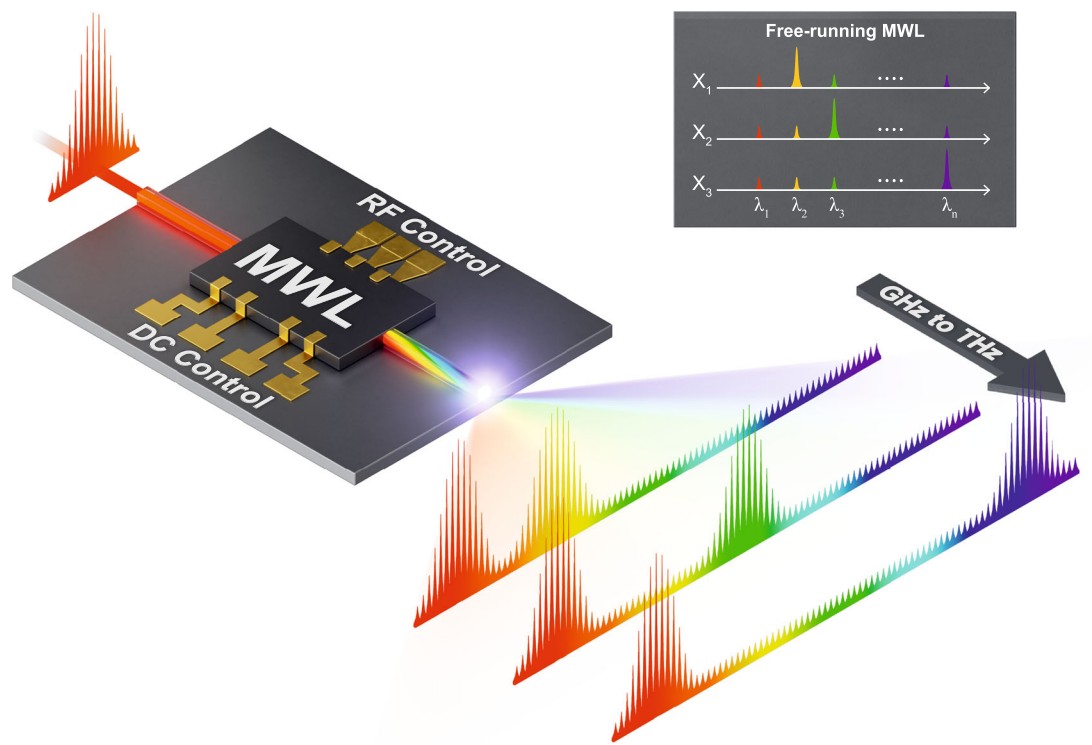

**Fig. 1 | THz-range frequency comb spectral multiplication with a multi-wavelength laser (MWL).** A narrowband frequency comb is spectrally multiplied to frequency offsets ranging from tens of GHz to few THz by injecting it in a controllable integrated multi-wavelength laser (MWL) photonic chip. Without optical injection, the MWL can be controlled to emit at wavelengths $\lambda_1, \lambda_2, ..., \lambda_n$. High-speed mode switching at nanosecond time scales is achieved by adjusting a

single control parameter X. Under frequency comb injection, the inherent mode coupling in the MWL leads to frequency comb multiplication at around the MWL emission wavelengths. By adjusting the control parameter, switching between GHz-range and THz-range frequency comb spectral multiplication can be achieved at nanosecond time scales.

of a few GHz to tens of GHz, is injected around one of the suppressed modes of the MWL. Owing to the intrinsic coupling between the modes of the MWL, the injected comb is spectrally multiplied around the different wavelengths emitted by the MWL. Similarly, the control mechanism of the MWL allows for selecting around which wavelength(s) the spectral multiplication will take place, thereby modifying the envelope of the output comb. Due to the fast response of the laser, switching between different wavelengths can be achieved at the nanosecond timescale. Such a THz-range spectral multiplication of narrowband frequency combs allows for energy-efficient and programmable spectral control over a THz range, which can be adjusted at nanosecond (GHz) speeds advantageous for, e.g., spectroscopy applications, THz generation, or even flexible networks.

## Demonstration of comb multiplication up to 1.3 THz in a dual-cavity laser

The backbone of the proposed spectral multiplication technique is a multi-wavelength laser combined with the optical injection of a frequency comb. In this work, the MWL is a dual-cavity laser with two detuned distributed Bragg reflectors (DBR) placed in a row, a shared semiconductor optical amplifier (SOA), and a broadband reflector (BR), see Fig. 2a, and refer to the Methods for a detailed description of the photonic integrated circuit. The DBRs are detuned by

approximately 10 nm (1.3 THz) with corresponding Bragg wavelengths at approximately 1537 nm and 1547 nm. While the laser was initially optimized to emit at two distinct wavelengths with 10 nm separation, multiple longitudinal modes can emit around each DBR's Bragg wavelength depending on the current sent to the DBRs. In this work, we use the term "mode" to refer solely to these longitudinal modes. The laser threshold current is at 21 mA, and we operate at $I_{SOA1} = 30$ mA. In this configuration, see Methods, the free-running laser emits at $\lambda_4 = 1547.58$ nm, a longitudinal mode corresponding to the cavity defined by DBR$_2$. Three additional residual modes can be observed at $\lambda_1 = 1536.70$ nm, $\lambda_2 = 1536.92$ nm, and $\lambda_3 = 1537.13$ nm, corresponding to the cavity defined by DBR$_1$. These modes are suppressed by more than 30 dB compared to the power of the dominant mode, see Section 1 of the Supplementary Information. The optically injected signal consists of a frequency comb featuring 5 tones with an FSR of 1 GHz. Multi-wavelength emission requires a careful gain balance between all emitted modes, but, from this point of view, optical injection represents a major disturbance. Thus, to avoid switching off un-injected modes when the optical injection is added, we inject the comb around $\lambda_3$ that is strongly suppressed in the free-running configuration. This way we can achieve so-called partial-locking, i.e. the other modes are still on, despite sending a relatively strong injection signal[32]. When the injection signal is powerful enough and the detuning, defined as

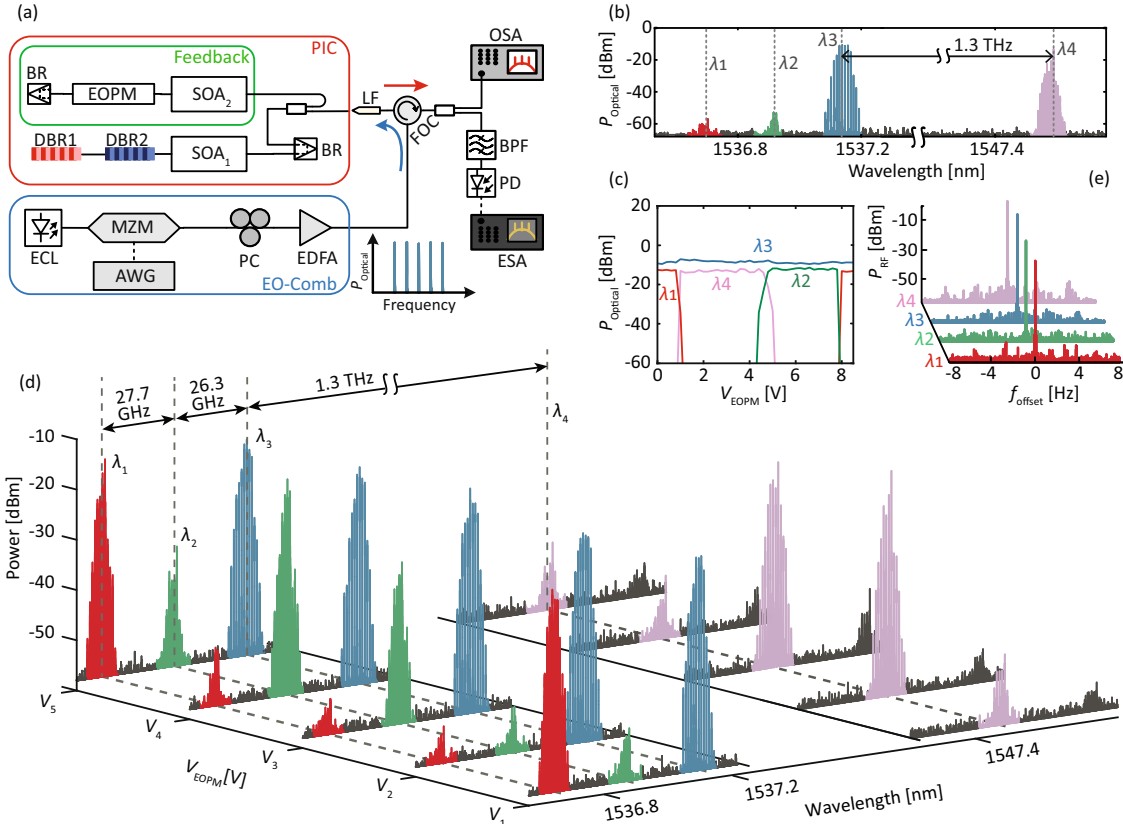

**Fig. 2 | THz-range agile frequency comb spectral multiplication using a feedback-controlled semiconductor MWL. a** Experimental setup. To generate the EO-comb, the light from an external cavity laser (ECL) is sent through a Mach-Zehnder modulator (MZM, $V_\pi \approx 6$ V) driven by an arbitrary waveform generator (AWG). The polarization and strength of this injected signal is adjusted with a polarization controller (PC) and an EDFA, respectively. The light is coupled in and out of the photonic integrated circuit (PIC) containing the MWL with feedback cavity via a lensed fiber (LF). A high-resolution (5 MHz RBW) optical spectrum analyzer (OSA) records the output spectrum. An optical bandpass filter (BPF) followed by a high-speed photodiode (40 GHz BW) is used to select a sub-comb which is then analyzed by an electrical spectrum analyzer (ESA). **b** Optical spectrum of the

re-generated and multiplied combs when the feedback is off ($I_{SOA2} = 0$ mA) and the narrowband comb is injected around $\lambda_3$. The dashed lines indicate the wavelengths $\lambda_1$ to $\lambda_4$ of the residual longitudinal modes of the MWL. Without feedback, the frequency comb is multiplied at around $\lambda_4$. **c** Optical power of the various sub-combs as a function of the voltage, $V_{EOPM}$, applied to the EOPM for $I_{SOA2} = 21$ mA and comb injection around $\lambda_3$. Comb switching occurs between the multiplied combs at around $\lambda_1$, $\lambda_2$, and $\lambda_4$. **d** Optical spectral for various $V_{EOPM}$ values: $V_1 = 0$ V, $V_2 = 3$ V, $V_3 = 4.6$ V, $V_4 = 7$ V, and $V_5 = 8$ V. **e** RF spectrum of the resulting beating between the tones of each comb (multiplied and re-generated). The low (< 1 Hz) linewidth, limited by the RBW of the ESA (1 Hz), shows that the coherence of the injected comb is preserved after the injection.

the frequency difference between the central peak of the comb and the injected wavelength, is small enough, the multi-wavelength laser re-generates the comb around the injected wavelength, see Fig. 2(b). We set the detuning and the injection strength to $\Delta = 2.3$ GHz and $\kappa_{inj} = 9.5$ dBm, respectively. Here, we define the injection strength as the optical power of the injected signal at the lensed fiber (LF) input. It is important to note that the injection strength and the detuning have a visible effect on the laser output, for more details see Section 2 of the Supplementary Information. With the frequency comb being picked up by $\lambda_3$, spectral multiplication of the modulated signal at $\lambda_4$ is obtained through the modulation of the carrier population. Because the injected mode experiences amplitude modulation induced by the frequency comb, the carrier population of the laser will experience similar fluctuations. Then, due to the carrier coupling between all modes of the MWL, frequency combs emerge at the other wavelengths emitted by the MWL, in this case, $\lambda_4$. We thus obtain multiplied combs separated by up to 1.3 THz. Depending on the laser configuration, we can multiply the frequency combs to different frequency offsets from a few GHz up to a few THz, depending on the mode emitted by the MWL. Although a similar behavior has already been reported in other laser systems[29], this is the first time, to the best of our knowledge, that this effect is triggered across such a large frequency offset.

## Frequency-agile spectral multiplication through optical feedback control

While spectral multiplication is interesting in itself, e.g., for spectral shaping of frequency combs, it also offers an exciting opportunity in terms of tunability which is a crucial point to enable versatile technological solutions. Here, we rely on weak optical feedback from a monolithically integrated feedback cavity that simply comprises a semiconductor optical amplifier (SOA) to control the feedback strength and an electro-optic phase modulator (EOPM) to control the feedback phase, see Fig. 2(a). In previous work, we showed controllable switching between different wavelengths emitted by our MWL[33]. The switching mechanism is primarily based on a careful design of the feedback cavity length, which should ensure that the different modes are sent back towards the laser in anti-phase, roughly acting like a Fabry-Perot interferometer. Tuning the feedback phase then allows us to select which wavelength is resonant in the external cavity. This approach works well to control more than two modes in the laser cavity[34], even with a design initially optimized for two specific wavelengths. The feedback phase is controlled by applying a voltage $V_{EOPM}$ to the EOPM ($V_\pi \approx 12$ V) of the feedback cavity. We set the feedback strength by driving SOA2 with $I_{SOA2} = 21$ mA, and keep the injection current of the MWL at $I_{SOA1} = 30$ mA, as previously. Without optical injection, such feedback control enables switching between three modes of the MWL at $\lambda_1$, $\lambda_2$, and $\lambda_4$. In this case, the mode at $\lambda_3$ remains strongly suppressed for any $V_{EOPM}$ value, which is why we inject the frequency comb around $\lambda_3$. When EO-comb injection is present, we observe similar switching performances. As a result, by simply tuning the phase of the optical feedback cavity we can switch between three different multiplied combs. The re-generated comb which is emitted around the injected mode always remains present, see Fig. 2(c), which shows the evolution of the power for each comb when $V_{EOPM}$ is varied from 0 V to 8.5 V. Below 1 V, the comb is multiplied to the first longitudinal mode of the first cavity at $\lambda_1$. Close to 1 V, the comb at $\lambda_4$ starts emitting leading to comb switching from $\lambda_1$ to $\lambda_4$. For $V_{EOPM} > 5$ V, emission around the second longitudinal mode of the first cavity, at $\lambda_2$, is gradually switched on while emission at $\lambda_4$ is turned off. While the first transition from $\lambda_1$ to $\lambda_4$ occurring for $V_{EOPM} \approx 1$ V appears to be rather sharp, the second transition from $\lambda_4$ to $\lambda_2$ is significantly smoother with the two combs appearing simultaneously in a small range of voltages around $V_{EOPM} \approx 5$ V. Note that, without injection, the laser exhibits similar switching behavior between $\lambda_1$, $\lambda_2$, and $\lambda_4$, see Section 1 of the Supplementary Information. Figure 2(d) depicts

the spectral multiplication of the injected frequency comb at different $V_{EOPM}$ values. In this MWL configuration, we achieve sequential emission of three multiplied combs in the mm-wave and THz range with an FSR of 1 GHz and central frequencies around 26 GHz, 54 GHz, and 1.3 THz, as shown in cases $V_4$, $V_1$, and $V_2$ in Fig. 2(d), respectively. We achieve an extinction ratio, i.e., power difference between the on and off state, for each multiplied comb, of above 30 dB, only relying on phase-controlled optical feedback. The peak power of the multiplied combs, measured just after the lensed fiber, is between -12 dBm and -13 dBm, comparable to the optical power of -11 dBm of the comb generated around the injected wavelength. With careful tuning of the feedback phase, the simultaneous emission of two multiplied combs, e.g., at $\lambda_2$ and $\lambda_4$, can be achieved, see case $V_3$ in Fig. 2(d). While changing $V_{EOPM}$, we observe that the center wavelength of both the re-generated and multiplied comb remains stable with variations of $\pm 100$ MHz for the multiplied combs, based on optical spectra measurements with limited absolute accuracy. Similarly, the comb bandwidth and shape are overall well preserved with only small variations. In particular, we measure bandwidths of up to 10 GHz for both the re-generated and multiplied combs. We can therefore conclude that, besides the wavelength switching, the impact of the feedback on the comb generation process is limited. This is an important point as it suggests that the two processes - optical injection and optical feedback - could potentially be analyzed and optimized rather independently.

An important added value of this control technique is the speed. The EOPM has a large bandwidth of up to several tens of GHz even on generic foundry platforms[35]. Besides, the laser is effectively switching between two different states which are both above the threshold: the carrier population is already filled in and the feedback only leads to small gain variations. As a result, even without proper RF optimization, we achieve switching times below 4 ns, further details are given in Section 4 of the Supplementary Information. We believe that these performances could be further improved by using co-planar waveguide transmission lines and electrical isolation of the EOPM. Regarding the characteristics of the sub-combs, we observe that, while the re-generated and multiplied combs feature different bandwidths depending on the MWL and injection parameters, the RF linewidth of the original EO-comb is preserved. We confirm it experimentally by isolating sub-combs using an optical filter and analyzing their respective RF spectrum using a photodiode (PD) and an electrical spectrum analyzer (ESA). Fig. 2e depicts the resulting tone centered at 1 GHz featuring a sub-Hz linewidth, limited by the ESA resolution bandwidth. There is, however, only a limited phase correlation between the different combs emerging at different wavelengths. We confirm it by isolating the two neighboring sub-combs for $V_{EOPM} = 7$ V, at around $\lambda_2$ and $\lambda_3$, see Fig. 3a. The resulting RF spectrum, Fig. 3b, features a broad beatnote at around 25 GHz with a long-term 3-dB-linewidth of 38 MHz, indicative of the lack of coherence between the sub-combs. In the following section, we further discuss how to achieve phase correlation between the various sub-combs.

## Cascaded phase-locking of multiplied combs

Unlike the tones of each re-generated or multiplied comb, the different wavelengths ($\lambda_1, \lambda_2, ..., \lambda_n$) emitted by the MWL are not inherently phase-locked nor phase-correlated. This means that the different sub-combs are actually not phase-correlated with one another. The RF linewidth of the tones obtained from the beating of two sub-combs is several MHz, matching the typical laser linewidth of the MWL. Yet, various techniques could be employed to phase-correlated the sub-combs, e.g., exploiting four-wave mixing (FWM) effects in SOA[36], thereby achieving comb broadening. We discuss here two different approaches leading to the phase-locking of the multiplied combs. The first one consists of adding an extra tone to the modulated optical injection on top of the EO comb already present. The extra tone is directly added using the AWG, see Fig. 4a–c. When this extra tone is

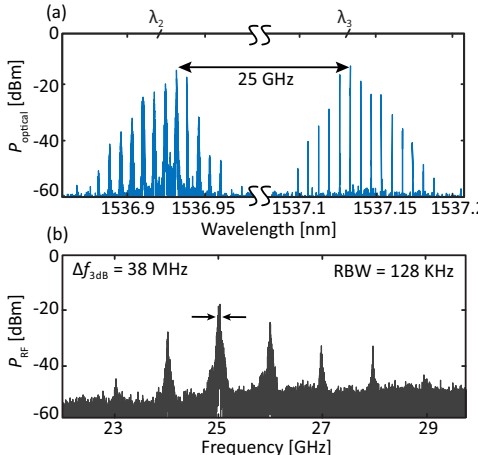

**Fig. 3 | Phase correlation between sub-combs. a** Optical spectrum of the sub-combs centered at around $\lambda_2$ and $\lambda_3$. The large difference in optical linewidth between the tones of the two sub-combs is indicative of the lack of coherence between the sub-combs. **b** RF spectrum of the photomixed sub-combs. The strongest beatnote features a long-term 3-dB-linewidth of approximately 38 MHz, linked to the absence of coherence between the sub-combs.

close to an active mode of the MWL, the multiplied comb can then be actively phase-locked to the re-generated comb. This approach is illustrated in Fig. 4d–f considering the same laser configuration and feedback detailed in the previous subsection, though we only focus on the longitudinal modes belonging to the laser cavity defined by DBR₁. Without injection and with $I_{SOA2} = 21$ mA and $V_{EOPM} = 0$ V, $\lambda_1$ is dominant and the other longitudinal modes, $\lambda_2$, and $\lambda_3$ are strongly suppressed as shown in Fig. 4d. The EO-comb signal is injected around $\lambda_3$, while the extra tone which is phase-locked with the EO-comb is injected around $\lambda_2$, at an offset frequency $f_{tone}$ from the center of the comb. The EO-comb injection triggers the OFC re-generation at wavelength $\lambda_3$ while the extra tone triggers a multiplied comb around $\lambda_2$, see Fig. 4e. We isolate the two combs emerging around $\lambda_2$ and $\lambda_3$ with a bandpass filter (BPF-I) and impinge the light into a PD followed by an ESA. We record the resulting spectrum with the ESA and observe an RF comb centered at around 26 GHz with the associated tones featuring linewidths below 1 Hz confirming an excellent phase correlation. Adding an extra tone in the injected comb, therefore, appears to be an effective way to force phase-locking between the re-generated and neighboring multiplied combs. It can however quickly become challenging as the wavelength difference increases and could even be out of reach for frequency differences in the THz as demonstrated earlier in this paper.

On the other hand, we also show that a careful tuning of the frequency $f_{tone}$ of the extra tone can trigger the emission of a second

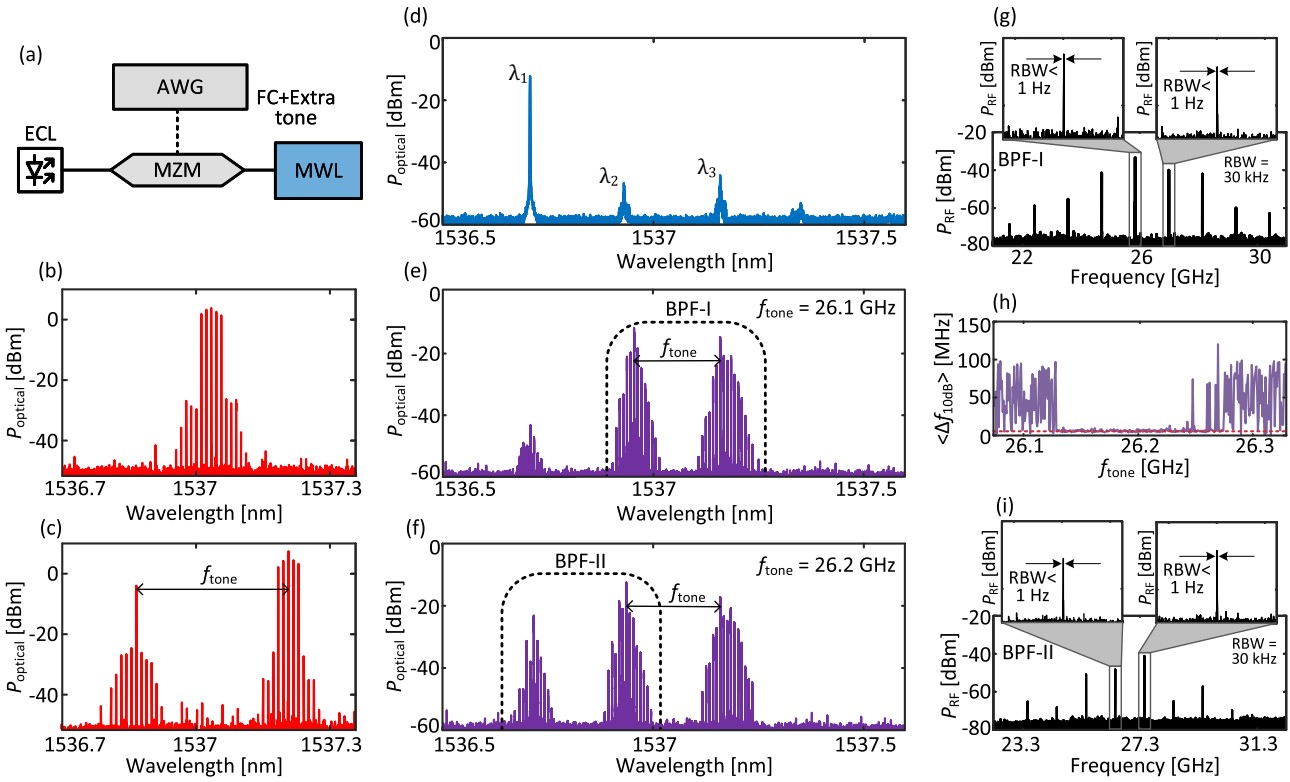

**Fig. 4 | Coherence control through cascaded phase locking. a** Experimental setup: A narrowband optical frequency comb with an extra tone is generated by modulating the light from an external-cavity laser (ECL) with a Mach-Zehnder modulator (MZM) driven by an arbitrary waveform generator (AWG). **b** Optical spectrum of the narrowband frequency comb with 1 GHz FSR. **c** Optical spectrum of the combined narrowband 1-GHz-FSR comb and side tone at a frequency difference $f_{tone}$ with respect to the center of the comb. **d** Optical spectrum, before injection, of the longitudinal modes associated with the long cavity of the MWL defined by DBR₁. **e** Optical spectrum of the MWL output under comb injection around $\lambda_3$ and $f_{tone} = 26.1$ GHz. The spectrum shows the re-generated and multiplied combs at around $\lambda_3$ and $\lambda_2$, respectively. At such $f_{tone}$ value, the extra tone is injected around $\lambda_2$, leading to comb multiplication around $\lambda_2$. **f** Optical spectrum of the re-generated and multiplied combs at well-tuned $f_{tone}$ of 26.2 GHz. Accurate optimization of the offset of the extra tone leads to cascaded phase locking of the multiplied comb emitting around $\lambda_1$. **g** RF spectrum resulting from the beating of the re-generated and multiplied comb for $f_{tone} = 26.1$ GHz. The inset indicates the narrow linewidth (<1 Hz) of the beating at the highest resolution bandwidth (RBW) of our ESA (1 Hz). **h** Averaged optical linewidth, measured at 10 dB, of the tones from the comb at $\lambda_1$ as a function of $f_{tone}$. The red dashed line at 5 MHz depicts the RBW limitation of the OSA. **i** RF spectrum resulting from the beating of the two multiplied combs for $f_{tone} = 26.2$ GHz. The inset indicates the narrow linewidth (<1 Hz) of the beating at the highest RBW of our ESA.

multiplied comb, phase-locked to the two other combs, around $\lambda_1$. When the frequency of the extra tone $f_{tone}$ is changed, the power, bandwidth, and linewidth of the comb emitting around $\lambda_1$ vary significantly. At well-optimized frequency offsets, the second multiplied comb at around $\lambda_1$ experiences a major linewidth reduction and power amplification. Figure 4(h), depicts the average value of the long-term 10-dB optical linewidth[37] of the comb lines emitting around $\lambda_1$, as a function of the $f_{tone}$, measured with an optical spectrum analyzer (OSA) with a resolution bandwidth of 5 MHz. The linewidth reduction occurs sharply for $f_{tone}$ values ranging from 26.14 GHz to 26.26 GHz. To accurately measure the phase correlation between the two multiplied combs around $\lambda_1$ and $\lambda_2$, we isolate them using a bandpass filter (BPF-II) and send the output to a PD followed by an ESA. Figure 4(i) depicts the resulting RF spectrum for $f_{tone} = 26.2$ GHz. The RF spectrum contains a frequency comb centered at the beat note frequency between the two OFCs, with each tone featuring a linewidth below the resolution limit of 1 Hz of our ESA. Outside the optimized range, the RF spectrum depicts broadband noise, see Section 5 of the Supplementary Information. These tones are generated from the beating between comb lines of the two multiplied combs, therefore, by fine-tuning the frequency offset of the extra tone we can achieve a phase correlation between the three neighboring combs. We were not able to observe a similar feature with the multiplied comb emitting around $\lambda_4$ with a 1.3 THz separation from the injected comb, with the current configuration between the modes of our MWL. One interpretation is linked to the interaction between the re-generated and phase-locked multiplied combs, which leads to a modulated comb appearing at their beating frequency. By tuning the frequency offset, $f_{tone}$, the resulting modulation caused by the beating can be locked on the first mode $\lambda_1$ which is not directly affected by the injection. In other words, the interaction between the two neighboring combs leads to a cascaded phase locking of the modes. We believe that cascaded phase-locking can advantageously be extended to additional modes leading to even larger frequency differences, although we could not find a suitable configuration to test it experimentally.

**Numerical model and simulations: impact of the mode coupling parameter**

Looking at the dynamical behavior of semiconductor lasers, rate equation models, despite their simplicity, have been shown to qualitatively predict most if not all, dynamical features reported experimentally. In addition, their simplicity allows the use of advanced techniques such as continuation to further clarify the nonlinear mechanisms at play. Here, we use a phenomenological multi-mode model initially derived for solid-state lasers but adapted for semiconductor lasers[38]. We have extended this model to include optical frequency comb injection and optical feedback to match our experimental scheme, see Methods. Considering the complexity of the coupling mechanism - represented here by the cross-saturation between each mode pair - and that the number of parameters describing such coupling quickly increases with the number of modes, we report here results obtained for two-mode model only, while in Section 7 of the Supplementary Information, we report results for the three-mode model. Using typical semiconductor laser parameters, comb multiplication can be easily obtained providing that the gain balance and mode coupling parameter between the different modes have been adjusted to allow for simultaneous emission of both the re-generated and multiplied combs[32]. Similarly, feedback-induced switching between the multiplied combs is well reproduced qualitatively as shown in Section 7 of the Supplementary Information. In fact, the main unknown is the value of the cross-saturation parameter $\beta$ and its impact on the comb generation mechanism. To bring new insight into this question, we have explored the parameter space for a two-mode model for different values of the mode coupling parameter $\beta$ as pictured in Fig. 5. An important aspect to consider is that cross-saturation influences the power balance between the modes. To keep

the same balance, which is crucial in the use case considered here, we need to carefully adjust the modal gain after changing the cross-saturation parameter[32]. We use this approach to analyze the effect of changing the coupling strength between modes while keeping all other parameters, except the modal gain, fixed. We adjust the parameters to obtain a 40 dB suppression ratio between the dominant and suppressed mode, similar to experimental observations. Next, we add an injection of an optical frequency comb with 5 tones to the suppressed mode, see Methods.

As already reported by others[32,39], we observe that the injection parameters have a strong impact on the comb generation process for both injected and un-injected modes. For positive detuning values, on the right side of the locking region in Fig. 5, spectral multiplication appears, exemplary spectra of both the re-generated and multiplied comb are depicted in panel (4) and (5) of Fig. 5. Note that the locking range is defined for CW optical injection and it is shown by red dashed line. Moreover, the power balance between the re-generated and multiplied combs can be easily adjusted by tuning either the injection strength or the detuning. For negative detuning values, we observe a small region showing broad comb bandwidth for both the re-generated and multiplied comb, but it is also where different dynamical behavior is typically observed. To estimate the FSR of the output signal, we consider the signal with spectral components surpassing a predefined threshold. Subsequently, we calculate the separation between the different spectral components of the output signals. In panel (3) of Fig. 5, the presence of black dots signifies areas where the separation between the spectral components of the output signal does not match the FSR of the injected comb, indicating different dynamical behavior for both modes, see Section 6 of the Supplementary Information. Although we did not perform such a systematic investigation experimentally, these features are well in line with our observations. Investigating the impact of the coupling parameter, we mainly see that the locking region is shrinking, pushed towards stronger injection when the coupling is decreased, see Fig. 5c and d. Different dynamical behaviors also appear to be partially inhibited when decreasing the coupling parameter. Interestingly, with respect to the locking range, the regions where spectral multiplication, suppression of the multiplied comb, and dynamics occur are preserved. That is, additional dynamics occur at negative detuning values, suppression of the multiplied comb occurs within the locking range, while spectral multiplication with large comb bandwidths occurs at the positive detuning side of the locking range. In particular, we foresee promising configurations emerging for weak injection and slightly negative detuning where much larger comb bandwidths appear. Our numerical investigations, therefore, suggest that weaker coupling between wavelengths might be desirable in the context of spectral multiplication. However, the wavelength coupling remains to be properly quantified experimentally, while its actual impact still needs to be investigated in detail.

## Discussion

We have shown the potential of multi-wavelength lasers to achieve spectral multiplication of optical frequency combs in a flexible and efficient way. We demonstrate spectral multiplication over more than 10 nm (1.3 THz) with adjustable frequency through a single control parameter. A key advantage of our approach is that it is scalable in frequency, with the fundamental limitation being the bandwidth of the gain medium. Yet, for the generic foundry InP integration platform used, lasers with a tuning range over 75 nm (approximately 10 THz at 1550 nm) have already been reported[40]. In addition, complete integration of our system on a single photonic integrated circuit is well within reach as commercial generic foundry platforms already provide phase modulators with more than 25 GHz bandwidth directly in their standard process (and even > 50 GHz for electro-absorption modulators[41]). Secondly, we confirm that tailored phase-controlled optical feedback can be efficiently used to control multi-wavelength

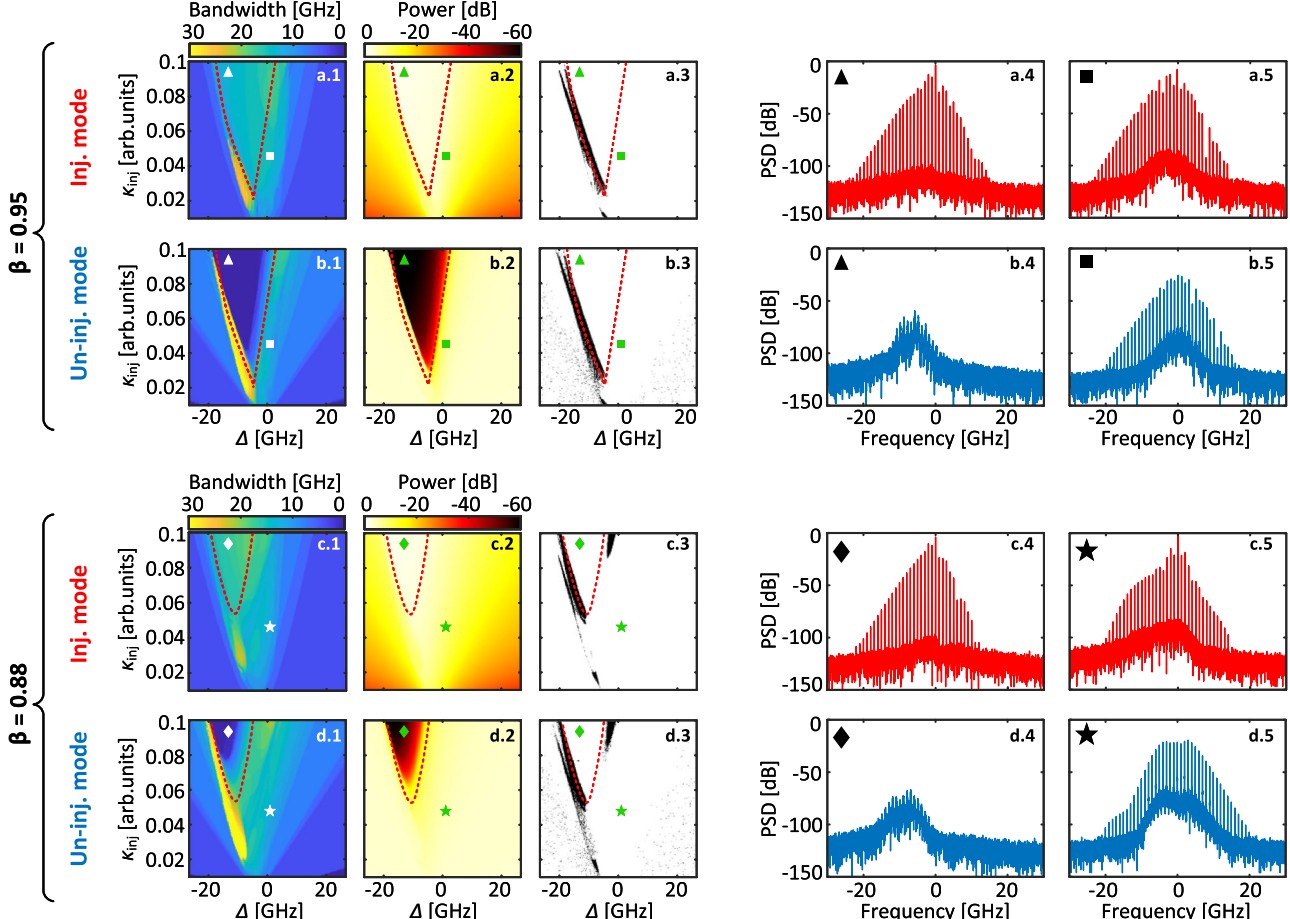

**Fig. 5 | Numerical investigation of the impact of the cross-coupling parameter $\beta$ on the frequency comb multiplication for a two-mode model, i.e., dual-wavelength laser.** The laser dynamics acquired for $\beta = 0.95$ are presented for the injected (**a**) and un-injected (**b**) modes. Similarly, the dynamics obtained for $\beta = 0.88$ are illustrated for the injected (**c**) and un-injected (**d**) modes. For each line, panel (1) displays the $\kappa_{inj}$-$\Delta$ map of the bandwidth of the output signal emerging around both the injected (**a**, **c**) and un-injected (**b**, **d**) modes. Panel (2) represents the $\kappa_{inj}$-$\Delta$ map of the total power of the output signal, while the black dots in panel (3) show where the FSR estimated from the output signal doesn't match the FSR of the injected comb, thus indicating a different type of dynamics. Spectra of the

output signal emerging around the injected (a.4, a.5) and un-injected (b.4, b.5) modes for two sets of $\kappa_{inj}$ and $\Delta$ values indicated by the triangles and the rectangles in the $\kappa_{inj}$-$\Delta$ maps where $\beta = 0.95$. The triangle is located within the locking range (shown by red dashed line), while the rectangle resides outside the locking range within the positive detuning values. The locking range is computed without modulation and for a CW injection. Spectra of the output signal emerging around the injected (c.4, c.5) and un-injected (d.4, d.5) modes for two sets of $\kappa_{inj}$ and $\Delta$ values at $\beta = 0.88$. The two sets of parameters are depicted by diamonds and stars within and outside the locking range, respectively.

lasers even under optical frequency comb injection. We demonstrate that weak feedback allows tuning the power balance between the different modes of the MWL without disrupting the internal mode competition on which we rely for spectral multiplication. In addition, unlike other schemes relying on strong feedback forcing to achieve switching - e.g., using internal frequency filters[42] or the so-called extended cavity regime[31,43] – we expect to have a significantly lower requirement in terms of power for the optical injection signal. Finally, while there is no phase-locking between the different wavelengths of our MWL – only a noise correlation already identified as an important added value for high-frequency signal generation[44] – we show that the addition of an extra tone in the injected EO-comb can trigger a cascaded mechanism leading to phase-locking of neighboring longitudinal modes. From an application viewpoint, it could benefit all use cases of frequency combs for which additional flexibility and programmability of the comb parameters are needed, but also applications where the extension of high-quality combs towards THz frequency is required. Our results show that it is possible to control the emission of the MWL thereby enabling THz-range frequency comb multiplication. When combined with THz photomixers, these chip-scale devices would enable compact, low-cost sources for programmable THz comb generation which

could allow for ultra-high speed wireless communications (for 6G and beyond) and spectroscopy of biomolecules. Alternatively, the proposed spectral multiplication scheme could bridge the gap between narrowband low-FSR combs[45,46] and broadband large-FSR combs[28,47]. Indeed, our approach has the potential to drastically extend the current capability to select the best of both worlds by copying the properties of the dense comb while maintaining the coherence provided by the broad comb.

## Methods

### Photonic integrated circuit design: dual-cavity laser with phase-controlled optical feedback

The multi-wavelength lasers (MWL) monolithically integrated with the optical feedback cavity have been fabricated on the InP generic foundry platform of SMART Photonics[41]. The detailed structure is schematically described in Fig. 2a. The design only uses the standard building blocks of the platform. The laser is a dual-cavity laser composed of a broadband two-port multimode interference reflector (BR, 40% reflectivity), a 500 $\mu$m-long semiconductor optical amplifier (SOA), and two distributed Bragg reflectors (DBRs) with a length of $L_1 = 500\,\mu$m and $L_2 = 250\,\mu$m, respectively. The DBRs are arranged

sequentially and act as wavelength-selective elements. The pitch of the DBRs is set to 0.236 $\mu m$ and 0.234 $\mu m$ for $DBR_1$ and $DBR_2$ respectively, leading to roughly 10 nm spectral separation between the two cavity modes. The laser consists of two cavities with a length of 1396.26 $\mu m$ and 1082.43 $\mu m$ corresponding to $DBR_1$ and $DBR_2$, respectively. The MWL is coupled to an external cavity via an 85/15 multimode inter-ference (MMI) splitter, meaning that only 15% of the light is sent toward the feedback cavity. The external cavity is composed of a 300 $\mu m$-long SOA, an electro-optic phase modulator (EOPM), and a one-port broadband reflector (BR) with 80% reflectivity. The total length of the external cavity is approximately 2.42 mm thus corresponding to the so-called "short-cavity regime". At SOA transparency, assuming all other components are ideal, we estimate that approximately 1.8% of the light is fed back toward the MWL cavity. The EOPM is designed to introduce a phase shift of $\pi$ when biased at approximately 12 V, i.e., $2\pi$ for a total round trip in the feedback cavity. Further details on the laser and cavity designs can be found in Ref. 33. Both MWL and OFB are implemented on the same chip which is electrically packaged with all-metal pads being wire-bonded to PCB boards. In addition, it has been glued on a Peltier element including a thermistor. During our experiments, the temperature of the chip is set to 22 °C.

## Experimental setup for comb multiplication

The injected electro-optic (EO) comb is generated by a high-quality tunable laser (Keysight N7776C) whose optical output is then modu-lated by a LiNbO$_3$ Mach-Zehnder modulator (MZM, IxBlue MXAN-LN-40, 40 GHz bandwidth, $V_\pi = 6$ V). Electric signal modulation is gener-ated by an arbitrary waveform generator (AWG, Keysight M8194A) and sent to the RF port of the MZM. The comb characteristics such as the number of lines and free-spectral range are defined at the AWG level. The polarization controller (PC) is used to align the laser polarization with the output of the modulator. The EO-comb is then amplified using an erbium-doped fiber amplifier (EDFA) to boost the comb's optical power. To couple the light in and out of the photonic integrated circuit containing the MWL and the optical feedback cavity, we use a lensed fiber whose output is coupled to a fiber optic circulator (FOC) to avoid back reflections. The MWL output is then sent towards a high-resolution optical spectrum analyzer (APEX, AP2083, resolution down to 5 MHz or 40 fm), and a large bandwidth photodetector (Thorlabs, RXM42AF 42 GHz) coupled to an RF spectrum analyzer (Keysight, MXA-N9021B). We set the current applied to $DBR_1$ and $DBR_2$ of the MWL to $I_{DBR1} = 0$ mA and $I_{DBR2} = 1$ mA, respectively. We define the injection strength by measuring the optical power of the injected signal before the lensed fiber. The detuning $\Delta$ is defined as the fre-quency difference between the frequency of the middle line of the comb $f_m$, and the free-running frequency of the mode under injection $f_s$: $\Delta = f_m - f_s$.

## Numerical model and simulations

We use the multimode rate equation model introduced by Viktorov et al.[38] derived from equations for solid-state lasers. Already normal-ized in time and extended to include the effect of optical feedback[48], we have added an extra term in the field equation to account for modulated optical injection[32]. By using the frequency of the master laser as reference frequency, we can remove the explicit time depen-dence thus making them an autonomous set of delayed-differential equations. These equations read as:

$$\frac{dF_1}{dt} = (1 + i\alpha)(g_1 N_1 - \frac{1 - g_1}{2})F_1$$
$$+ \kappa e^{-i\theta_1} F_1(t - \tau)e^{-i\Delta\tau} + \kappa_{inj}m(t) - i\Delta F_1, \quad (1)$$

$$\frac{dE_2}{dt} = (1 + i\alpha)(g_2 N_2 - \frac{1 - g_2}{2})E_2 + \kappa e^{-i\theta_2}E_2(t - \tau), \quad (2)$$

$$T\frac{dN_1}{dt} = P - N_1 - (1 + 2N_1)(g_1|E_1|^2 + g_2\beta|E_2|^2), \quad (3)$$

and

$$T\frac{dN_2}{dt} = P - N_2 - (1 + 2N_2)(\beta g_1|E_1|^2 + g_2|E_2|^2). \quad (4)$$

With $E_{1,2}$ being the normalized electrical field for modes 1 and 2 respectively, and $N_{1,2}$, the normalized carrier population density. All variables and parameters are normalized and further details of the normalization can be found in the original work[48]. It is however worth noting that all time/frequency parameters are normalized by the photon lifetime $\tau_p$. The linewidth enhancement factor is represented as $\alpha$ and the modal gains by $g_{1,2}$. $T$ is the the carrier lifetime (normalized by the photon lifetime) and $P$ is the normalized pump parameter. The mode coupling mechanism is driven by a parameter $\beta$ which models the cross-saturation. The optical feedback is introduced by the second terms of the field equations and is modeled by the feedback rate $\kappa$, the time-delay $\tau$ (normalized by the photon lifetime), and the feedback phase $\theta_{1,2}$. The latter depends on the wavelengths and is therefore different for modes 1 and 2. For the numerical results depicted in Fig. 5, optical feedback is not considered, i.e., $\kappa = 0$. The use of feedback as a control mechanism to switch between wavelengths is however demonstrated in the Supplementary Information, section 7. The optical injection is introduced in the last two terms of the first equation and modeled by the injection rate $\kappa_{inj}$, the modulation $m(t)$ (with the time parameter $t$ being normalized by the photon lifetime), and the detuning $\Delta$, i.e. the normalized frequency difference, between the master laser and the injected mode. The first equation is written for $F_1(t) = E_1(t)e^{-i\Delta t}$, i.e., the electrical field $E_1$ shifted in frequency by the detuning $\Delta$ to make the equation autonomous. We simulate the system using direct numerical integration with a Runge-Kutta 4th-order algorithm. In the simulation, the linewidth enhancement factor is set to $\alpha = 3$ and $T = 1000$. The pump parameter $P$ corresponds to a normalized laser injection current $J$ so that $P = (J - J_{th})/2J_{th}$, with $J_{th}$ being the threshold current. Here, we use $P = 0.5$, corresponding to an injection current 2 times the laser threshold. To fix the power difference between the two modes of the MWL, we need to adjust both gain and cross-saturation parameters. Here, we fix $g_1 = 0.995$ and consider two cases: 1) $g_2 = 0.827$ with $\beta_1 = 0.88$ and 2) $g_2 = 0.954$ with $\beta_2 = 0.95$. Both cases lead to a power difference of 40 dB between modes without injection or feedback. For numerical investigation as in the experiment, we inject a comb with five tones. To better compare it with the experimental observations, in Fig. 5 we have expressed both the FSR and the detuning values in Hz by assuming a photon lifetime of 3 ps. In particular, to simulate an FSR of about 1 GHz, we use a normalized FSR of 0.019 ($\approx 2\pi f \tau_p$ with $f = 1$ GHz and $\tau_p = 3$ ps). To obtain the bandwidth and the total power of the combs we define a threshold for the power of the comb lines at 30 dB below the power of the strongest comb line. Only the comb lines that appear above the threshold are considered for the bandwidth and power calculation. To evaluate the repetition rate, we compute the average spacing between the comb lines that appear above the threshold.

## Data availability

Experimental and numerical data generated in this study and pre-sented in this manuscript have been deposited in the Nature Figshare repository under accession code https://doi.org/10.6084/m9.figshare.23515581.

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

## Acknowledgements

This work was supported by the European Research Council (ERC, Starting Grant COLOR'UP 948129, MV), the Research Foundation Flanders (FWO, grants 1530318N, G0G0319N, MV, and postdoctoral fellowship grant 1275924N, PMP), the METHUSALEM program of the Flemish Government (Vlaamse Overheid).

## Author contributions

MV initiated, led, and supervised the project. The photonic IC including the MWL and feedback cavity has been designed in a previous project under the supervision of MV. PMP came up with the idea of the comb multiplication scheme. ML performed a detailed characterization of the MWL and identified suitable operating points. SA and PMP designed and built the setup. SA performed the experiments and simulations under the supervision of PMP. All authors discussed the results. SA, PMP, and MV wrote the manuscript while all authors helped revise it.

## Competing interests

The authors declare no competing interests.
