## [Peer Review File · Nature Communications]

REVIEWER COMMENTS

Reviewer #1 (Remarks to the Author):

The manuscript describes a method of spectral multiplication of a narrowband comb by optical injection to a multi-wavelength semiconductor laser. Demonstrating a proof-of-concept experimental result of 1.3 THz frequency offset, the authors claim that the method is able to cover a range of several THz, with the possibility of creating compact on-chip comb sources for THz applications. The manuscript is well written with clear details to support the method of optical feedback control and phase modulation via the multi-wavelength semiconductor laser adopted here to enable agile THz comb generation. The proposed method is believed to have the potential to extend the state-of-the-art on-chip combs generation based on micro-resonators or electro-optic modulators by enhancing the control lability as well as the spectral bandwidth.

Conclusively, the manuscript is strongly recommended for publication in NCOMMS as it is.

Reviewer #2 (Remarks to the Author):

Comments for the manuscript "Agile THz-range spectral multiplication of frequency combs using a multi-wavelength laser" (NCOMMS-23-25899) submitted to Nature Communications:

The authors have expanded the optical bandwidth of an electro-optic modulated optical frequency comb (EO comb) by means of injection locking into a multi-wavelength semiconductor laser (MWL) at 1540 nm. The EO comb is at 1537.13 nm (the bandwidth appears to be approximately 10 GHz, although not specified in the manuscript), and additional combs at 1536.70 nm, 1536.92 nm, and 1547.58 nm, in addition to the original 1537.13 nm, are made available by the injection-locked MWL. The center wavelength can be adjusted by controlling the feedback phase of the MWL (an electro-optic modulator in the MWL cavity), and the switching is as fast as 4 ns.

If the authors consider incorporating the following suggestions to improve their manuscript, it has the potential to be published in Nature Communications.

1. About the phase correlation between combs at different wavelengths:

As mentioned in the text (p. 5, l. 303-), combs at different wavelengths may not be phase-correlated unless an additional fast phase modulation at ~ 26 GHz is applied to the original EO comb. The authors

should provide experimental results to explicitly show evidence of the absence of the phase correlation without the 26 GHz modulation for comparison.

The authors mentioned that the comb at 1547.58 nm cannot be phase-correlated with others due to the large frequency separation (over 1 THz). This suggests that this system may not qualify as a "comb," which should be phase-correlated over its entire bandwidth. This raises concerns about the accuracy of the manuscript's title, as the "comb" is not broadened over the THz range.

2. In the theoretical section starting from p. 6:

Overall, the content described in this section is difficult to understand. For example,

* I struggle to comprehend the model. While the authors mention "two- and three-mode models," there seems to be no description of the "three-mode model." Additionally, I find it difficult to understand what is meant by a "mode," whether it refers to a single-frequency mode or a series of comb teeth within a single frequency bandwidth.

* The authors mention "spectral multiplication appears" in Fig. 4 (p. 7, l. 437), but I cannot discern how it is evident from the figure.

* Similarly, I cannot find clear evidence supporting the description in p. 7, l. 455 ("Interestingly, the region where spectral multiplication occurs is mostly preserved").

* Details about how the parameter values are normalized should be provided.

3. Optical spectra of the EO comb with and without the additional modulation at ~26 GHz for phase locking should be provided. This would clarify whether the MWL acts as a slave laser rather than an optical amplifier in the phase-correlated case.

4. The authors mention from p. 5, l. 377- about "cascaded phase locking." Although I don't fully understand it, I interpret it as phase locking over the full optical bandwidth of the comb being induced by the phase locking between two neighboring bands (e.g., the phase locking between λ_2 and λ_3 due to additional modulation of the EO comb causes phase locking with λ_1). However, the theoretical model does not seem to account for coherent interactions between the two modes. In fact, in the four equations on p. 9, the interaction between E1 and E2 depends on the intensity $|E1|^2$ and $|E2|^2$. Therefore, according to the model, it remains unclear why the phase correlation extends to modes where there are no comb teeth in the EO comb.

Additionally, the authors mention that they cannot "find a suitable configuration to test" the phase correlation between the extended modes. I suggest that such testing can be performed using a broadband Er fiber mode-locked laser comb.

Submission of revised version of Nature communication Manuscript Number NCOMMS-23-25899

“Agile THz-range spectral multiplication of frequency combs using a multi-wavelength laser”

Reviewer 1

The manuscript describes a method of spectral multiplication of a narrowband comb by optical injection to a multi-wavelength semiconductor laser. Demonstrating a proof-of-concept experimental result of 1.3 THz frequency offset, the authors claim that the method is able to cover a range of several THz, with the possibility of creating compact on-chip comb sources for THz applications. The manuscript is well written with clear details to support the method of optical feedback control and phase modulation via the multi-wavelength semiconductor laser adopted here to enable agile THz comb generation. The proposed method is believed to have the potential to extend the state-of-the-art on-chip combs generation based on micro-resonators or electro-optic modulators by enhancing the controllability as well as the spectral bandwidth.

Conclusively, the manuscript is strongly recommended for publication in NCOMMS as it is.

R: We would like to thank the reviewer for the favorable assessment of our work and the highly supportive recommendation.

Reviewer 2

The authors have expanded the optical bandwidth of an electro-optic modulated optical frequency comb (EO comb) by means of injection locking into a multi-wavelength semiconductor laser (MWL) at 1540 nm. The EO comb is at 1537.13 nm (the bandwidth appears to be approximately 10 GHz, although not specified in the manuscript), and additional combs at 1536.70 nm, 1536.92 nm, and 1547.58 nm, in addition to the original 1537.13 nm, are made available by the injection-locked MWL. The center wavelength can be adjusted by controlling the feedback phase of the MWL (an electro-optic modulator in the MWL cavity), and the switching is as fast as 4 ns.

If the authors consider incorporating the following suggestions to improve their manuscript, it has the potential to be published in Nature Communications.

R: We thank the reviewer for the overall positive evaluation and appreciate her/his comments regarding the content of our manuscript.

As pointed out by the reviewer, the useful bandwidth of both the re-generated and multiplied sub-combs reaches up to 10 GHz though it was, indeed, not specified in the manuscript.

Action taken: We have now explicitly mentioned the optical bandwidth of the re-generated and multiplied sub-combs in the manuscript (now line 282):

“... bandwidth and shape are overall well preserved with only small variations. In particular, we measure bandwidths of up to 10 GHz for both the re-generated and multiplied combs.”

- 1) About the phase correlation between combs at different wavelengths:
- As mentioned in the text (p. 5, l. 303-), combs at different wavelengths may not be phase-correlated unless an additional fast phase modulation at ~ 26 GHz is applied to the original EO comb. The authors should provide experimental results to explicitly show evidence of the absence of the phase correlation without the 26 GHz modulation for comparison.

R: This is, indeed, an important point: the different wavelengths of our multi-wavelength lasers are not, in general, phase correlated with or without injection, and thus the sub-combs are not phase correlated. As a result, the extra modulation tone (here at 26 GHz) is necessary to achieve coherence between the two combs (sub-combs) at different wavelengths, i.e., in this particular case, t_2 and t_3 , respectively. We agree that the absence of phase correlation could be highlighted more prominently to better show why this extra phase locking step needs to be taken.

Action taken: Following the reviewer’s suggestion, to explicitly show evidence of the lack of phase correlation between these 2 sub-combs, we have now incorporated an additional figure, namely Fig 3, into the revised manuscript, see below. The panel (a) in this figure illustrates the optical spectrum of the neighboring sub-combs centered at t_2 and t_3 emitted when setting $V_{EOPM} = 7$ V, while panel (b) depicts the resulting beatnotes after photomixing the two sub-combs. The observed distinction in optical linewidths between the regenerated and multiplied comb and the corresponding broad beatnote are indicative of the absence of phase correlation between these two sub-combs.

Fig. 3. Phase correlation between sub-combs. (a) Optical spectrum of the sub-combs centered at around t_2 and t_3 . The large difference in optical linewidth between the tones of the two sub-combs is indicative of the lack of coherence between the sub-combs. (b) RF spectrum of the photomixed sub-combs. The strongest beatnote features a long-term 3-dB-linewidth of approximately 38 MHz, linked to the absence of coherence between the sub-combs.

We have also added the link to this figure in the manuscript (line 315):

“... There is, however, only a limited phase correlation between the different combs emerging at different wavelengths. We confirm it by isolating the two neighbouring sub-combs for $V_{EOPM} = 7$ V, at around t_2 and t_3 , see Fig. 3(a). The resulting RF spectrum, , Fig. 3(b), features a broad beatnote at around 25 GHz with a long-term 3-dB-linewidth of 38 MHz, indicative of the lack of coherence between the sub-combs. In the following section, we further discuss how to achieve phase correlation between the various sub-combs.”

- b) The authors mentioned that the comb at 1547.58 nm cannot be phase-correlated with others due to the large frequency separation (over 1 THz). This suggests that this system may not qualify as a "comb," which should be phase-correlated over its entire bandwidth. This raises concerns about the accuracy of the manuscript's title, as the "comb" is not broadened over the THz range.

R: We agree with the reviewer in that the injected comb is not, strictly speaking, broadened over the 1 THz bandwidth, because the various sub-combs are not phase correlated. For this reason, we have refrained from using the term “comb broadening” and instead we use the term “spectral multiplication” to highlight the fact that each of the multiplied sub-combs preserves the inherent phase correlation between their comb lines. In other words, each sub-comb (regenerated and multiplied combs) is, by definition, a frequency comb. However, the spectrum of the light emitted by the multi-wavelength laser cannot be, as a whole, considered as a frequency comb because of the lack of phase correlation between the sub-combs.

With the same reasoning, we purposely chose the term “spectral multiplication” for the manuscript’s title to avoid ambiguity.

Action taken: to make the disclaimer more explicit, we have clarified what we mean with the term “comb spectral multiplication” and how it differs from “comb broadening” (line 88):

“Through optical injection of a narrowband comb, we achieve comb multiplication from tens of GHz up to 1.3 THz, while preserving the RF coherence of the injected comb, i.e. the phase correlation between its comb lines. However, phase correlation between the various sub-combs is not achieved per se. As such, we describe this process as spectral multiplication rather than comb broadening.”

For the record, we can find various techniques in the literature achieving phase correlation between different frequency combs which may be considered to overcome this current limitation of the proposed scheme; for example, by exploiting four-wave mixing (FWM) or carrier density modulation in semiconductor optical amplifiers (SOA) [P. D. Lakshmi Jayasimha, et. al., Opt. Express 27, 16560-16570 (2019)]. These investigations, however, go beyond the scope of our study, although they might be considered in further research. We have thus added the following sentence in the manuscript (line 331):

“Yet, various techniques could be employed to phase-correlate the sub-combs, e.g., exploiting four-wave mixing (FWM) effects in semiconductor optical amplifiers (SOA) [36], thereby achieving comb broadening.”

- 2) In the theoretical section starting from p. 6:

Overall, the content described in this section is difficult to understand. For example,

- a) I struggle to comprehend the model. While the authors mention "two- and three-mode models," there seems to be no description of the "three-mode model."

R: The reviewer is correct and we agree that the description of the model could lead to misunderstanding. In the original manuscript we indeed mentioned in the text “two- and three-mode models” (previously line 412), however in the main paper we only show the numerical results regarding the two-mode model because this is sufficient to highlight spectral multiplication. The numerical results concerning the three-mode model are only present in Section 7 of the SI where we report switching between the un-injected modes by modulating the optical feedback phase.

Action taken: To avoid confusion, we now only refer to the two-mode model in the theoretical section which is the underlying model used to obtain the results depicted in Fig. 4 (now Fig. 5). This model is enough to investigate the properties of the multiplied comb.

Action taken in SI: We have now included the explicit equations of the three-mode model in Section 7 of the SI.

- b) Additionally, I find it difficult to understand what is meant by a "mode," whether it refers to a single-frequency mode or a series of comb teeth within a single frequency bandwidth.

R: We agree that the term mode is ubiquitous and its meaning can vary significantly. In this paper, we use the term “mode” to solely refer to the cavity modes (or longitudinal modes) of the MWL. The wavelength associated with each mode is what we label as λ_1 to λ_4 . In this way, each sub-comb (consisting of a series of comb teeth within a single frequency bandwidth) is emitted around the wavelength of one of the cavity modes of the MWL.

Action taken: we have now clarified what “mode” refers to early on in the manuscript (line 158). which should be clearly distinguishable from the comb teeth of each sub-comb. In addition, we have also carefully reviewed the paper and confirm that all instances of the term “mode” are consistent.

“multiple longitudinal modes can emit around each DBR's Bragg wavelength depending on the current sent to the DBRs. In this work, we use the term "mode" to refer solely to these longitudinal modes.”

- c) The authors mention "spectral multiplication appears" in Fig. 4 (p. 7, l. 437), but I cannot discern how it is evident from the figure.

R: From Fig.4 alone, we agree that it is difficult to discern the appearance of the multiplied comb. We thank the reviewer for pointing this out.

Action taken: For each of the two values of the cross-saturation parameter β that we considered in Fig. 4 (now Fig. 5), we have now explicitly considered two sets of k_{inj} and Δ values and displayed the corresponding power spectra for both the regenerated and multiplied combs. These plots are displayed in Fig.5 (previous Fig. 4). With these added figures, it becomes evident that spectral multiplication appears for slightly positive detuning values (I and III), while the multiplied comb is mostly suppressed within the locking region (II and IV).

Fig. 5. Numerical investigation of the impact of the cross-coupling parameter β on the frequency comb multiplication for a two-mode model, i.e., dual-wavelength laser. (a) K_{inj} - Δ maps of the comb bandwidth, total power, and average FSR of the re-generated (top) and multiplied (bottom) combs for a $\beta = 0.95$. The re-generated comb shows strong power and large bandwidth at low detuning values. On the other hand, the multiplied comb shows reduction in both power and bandwidth roughly inside the locking range (shown by the red dashed line). The black dots in the right-most map represent the dynamics that appeared around both injected and un-injected modes. These dynamics are identified based on the average FSR. (b) Spectra of the re-generated (top) and multiplied (bottom) combs for two sets of K_{inj} and Δ values, indicated in panel (a) with the Roman numbers I and II. (c) K_{inj} - Δ maps of the comb bandwidth, total power, and average FSR of the re-generated (top) and multiplied (bottom) combs for a $\beta = 0.88$. Both re-generated and multiplied combs show a reasonable amount of power and bandwidth for a large range of K_{inj} and Δ values. In a small part of the locking range, i.e., higher injection strength, the multiplied comb is suppressed in power and bandwidth. (d) Spectra of the regenerated (top) and multiplied (bottom) combs for two sets of K_{inj} and Δ values, indicated in panel (c) with the Roman numbers III and IV.

- d) Similarly, I cannot find clear evidence supporting the description in p. 7, l. 455 ("Interestingly, the region where spectral multiplication occurs is mostly preserved").

R: With the statement "Interestingly, the region where spectral multiplication occurs is mostly preserved", we mean that for different cross-coupling parameters, the regions where dynamics, spectral multiplication or suppression of the multiplied comb occur are preserved with respect to the locking region. In particular, complex dynamics occur at negative detunings, suppression of the multiplied comb occurs within the locking range, while spectral multiplication with large bandwidth occurs at the positive detuning side of the locking range.

Action taken: we have rephrased the aforementioned statement in line 486:

“Interestingly, with respect to the locking range, the regions where spectral multiplication, suppression of the multiplied comb, and complex dynamics occur are preserved. that is, complex dynamics occur at negative detuning values, suppression of the multiplied comb occurs within the locking range, while spectral multiplication with large comb bandwidths occurs at the positive detuning side of the locking range.”

e) Details about how the parameter values are normalized should be provided.

R: We thank the reviewer for pointing this out. In Methods, we now refer explicitly to the publication (Ref. [48], Section III) which describes the normalization we use in our model. In addition, we indicate where we consider the normalization with respect to the photon lifetime, see the Section “Numerical model and simulations” in Methods.

3) Optical spectra of the EO comb with and without the additional modulation at ~26 GHz for phase locking should be provided. This would clarify whether the MWL acts as a slave laser rather than an optical amplifier in the phase-correlated case.

R: In the revised manuscript, we have now added panel (b) to Figure 4 (former Fig. 3) to illustrate the optical spectrum of the EO-comb also without the side tone.

(b) Optical spectrum of the narrowband frequency comb with 1 GHz FSR

With regards to the role of the MWL, i.e., whether it acts as a slave laser or optical amplifier, we believe that the distinction is difficult to make. When adding the sidetone, the MWL will act as a slave laser whereby the injected side tone will be “amplified”, i.e., re-generated [Yu-Han Hung and Sheng-Kwang Hwang, Opt. Express 23.5 (2015): 6520-6532]. Therefore, we cannot really separate optical amplification from the slave laser behavior.

4) The authors mention from p. 5, l. 377- about “cascaded phase locking.” Although I don’t fully understand it, I interpret it as phase locking over the full optical bandwidth of the comb being induced by the phase locking between two neighboring bands (e.g., the phase locking between λ_2 and λ_3 due to additional modulation of the EO comb causes phase locking with λ_1). However, the theoretical model does not seem to account for coherent interactions between the two modes. In fact, in the four equations on p. 9, the interaction between E1 and E2 depends on the intensity $|E1|^2$ and $|E2|^2$. Therefore, according to the model, it remains unclear why the phase correlation extends to modes where there are no comb teeth in the EO comb.

R: We agree with the interpretation of the reviewer which corresponds to what we mean with the term “cascaded phase locking”. Then, it is also correct that our model does not consider the coherent interaction between the various modes of the MWL and, as such, cannot reproduce this “cascaded phase locking” behavior as the different modes of the model are only coupled through the carrier population.

The fact that the “cascaded phase locking” requires a fine tuning of the f_{tone} frequency is, for us, a clear indication that this mechanism rely on a coherent interaction between the different wavelengths. Indeed, the injected signal is carefully filtered to suppress all possible spurious signals, so we know that it is not coming either from the optical injection or the carrier population coupling. At this stage, we suppose that four-wave mixing could potentially play a role. However, confirming this hypothesis experimentally or theoretically represents a significant challenge in the short term. This will nevertheless be a focus point for our future investigations.

Additionally, the authors mention that they cannot "find a suitable configuration to test" the phase correlation between the extended modes. I suggest that such testing can be performed using a broadband Er fiber mode-locked laser comb.

R: We thank the reviewer for this interesting suggestion, this could indeed be a practical solution and would remove the need to use “cascaded phase-locking” (at least in part). At this time, however, we unfortunately do not have access to such mode-locked laser, but we will certainly explore this possibility further.

REVIEWERS' COMMENTS

Reviewer #2 (Remarks to the Author):

Comments for the revised manuscript "Agile THz-range spectral multiplication of frequency combs using a multi-wavelength laser" (NCOMMS-23-25899) submitted to Nature Communications:

The manuscript has undergone significant revisions, and I believe it has the potential to be accepted for publication in Nature Communications, if the authors consider the suggested comments for further improvement. I regret not bringing some of these comments to the authors' attention in the 1st review. I apologize for overlooking them.

1. Fig. 5, (a) and (c): The letters of "Re-gen." and "Multi." in the left panels are not clear.
2. Fig. 5, (a) and (c), lines 5-6 in the caption of Fig. 5, p. 8, left column, line 3 in the text: The authors should explain what "complex dynamics" means. Correspondingly, detailed explanation of the black dots in the right panels of Fig. 5(a) and (c) is required. In addition, I cannot understand " $FSR=1$, $FSR=1$ " in the right panel.
3. Other minor comments:
 - i. p. 1, left, l. 16: "tunability. Meaning" -> "tunability, meaning"
 - ii. Fig. 2(d), the label of the axis: "Wavelenght" -> "Wavelength"
 - iii. Fig. 5, l. 5 in the caption: "dashed line" -> "solid line" (?)
 - iv. p. 8, left, l. 4: "that is," -> "That is,"

**Submission of revised Nature communication Manuscript Number
NCOMMS-23-25899**

“Agile THz-range spectral multiplication of frequency combs using a multi-wavelength laser”

Reviewer 2

The manuscript has undergone significant revisions, and I believe it has the potential to be accepted for publication in Nature Communications, if the authors consider the suggested comments for further improvement. I regret not bringing some of these comments to the authors' attention in the 1st review. I apologize for overlooking them.

R: We really appreciate the reviewer's last comments which, undoubtedly, will increase the quality of the publication.

- 1) Fig. 5, (a) and (c): The letters of “Re-gen.” and “Multi.” in the left panels are not clear.

R: To enhance clarity and avoid potential confusion in Figure 5, we have introduced the labels ‘Inj. mode’ and ‘Un-inj. mode’ on the left side of the respective panels. These panels are now designated as follows: for the injected and un-injected modes at $\bar{\beta} = 0.95$, they are labeled as (a.1 to a.6) and (b.1 to b.6), respectively. Likewise, the panels representing the injected and un-injected modes at $\bar{\beta} = 0.88$ are marked as (c.1 to c.6) and (d.1 to d.6), respectively. Additionally, we've organized the panels corresponding to $\bar{\beta} = 0.95$ and $\bar{\beta} = 0.88$ by enclosing them within braces for better visual grouping. We also replaced the usage of Roman number from the previous version with triangles, rectangles, diamonds, and stars to mark distinct points within the $\kappa_{inj}-\Delta$ maps. This alteration allows us to show examples of optical spectra for both injected and un-injected modes more effectively. The revised version of the figure and the corresponding revised caption are depicted below:

“Numerical investigation of the impact of the cross-coupling parameter β on the frequency comb multiplication for a two-mode model, i.e., dual-wavelength laser. The laser dynamics acquired for $\beta = 0.95$ are presented for the injected (a) and un-injected (b) modes. Similarly, the dynamics obtained for $\beta = 0.88$ are illustrated for the injected (c) and un-injected (d) modes. For each line, panel (1) displays the κ_{inj} - Δ map of the bandwidth of the output signal emerging around both the injected (a, c) and un-injected (b, d) modes. Panel (2) represents the κ_{inj} - Δ map of the total power of the output signal, while the black dots in panel (3) show where the FSR estimated from the output signal doesn't match the FSR of the injected comb, thus indicating a different type of dynamics. Spectra of the output signal emerging around the injected (a.4, a.5) and un-injected (b.4, b.5) modes for two sets of κ_{inj} and Δ values indicated by the triangles and the rectangles in the κ_{inj} - Δ maps where $\beta = 0.95$. The triangle is located within the locking range (shown by red dashed line), while the rectangle resides outside the locking range within the positive detuning values. The locking range is computed without modulation and for a CW injection. Spectra of the output signal emerging around the injected (c.4, c.5) and un-injected (d.4, d.5) modes for two sets of κ_{inj} and Δ values at $\beta = 0.88$. The two sets of parameters are depicted by diamonds and stars within and outside the locking range, respectively.”

- 2) Fig. 5, (a) and (c), lines 5-6 in the caption of Fig. 5, p. 8, left column, line 3 in the text: The authors should explain what "complex dynamics" means. Correspondingly, detailed explanation of the black dots in the right panels of Fig. 5(a) and (c) is required. In addition, I cannot understand "FSR=1, FSR=1" in the right panel.

R: Thank you for pointing this out. By complex dynamics we mean different dynamical behavior experienced by the optical field inside the laser cavity leading to either an output frequency comb whose FSR differs from that of the injected comb, or to other dynamics such as chaos or excitability. In this case, the terminology “complex dynamics” is indeed vague and ambiguous.

Action taken: Following the reviewer’s suggestion we now clarify what we mean by “complex dynamics”. We do so in the first occurrence of this term in our manuscript on page 7, right hand side, last paragraph.

“For negative detuning values, we observe a small region showing broad comb bandwidth for both the re-generated and multiplied comb, but it is also where *different dynamical behavior* is typically observed. To estimate the FSR of the output signal, we consider the signal with spectral components surpassing a predefined threshold. Subsequently, we calculate the separation between the different spectral components of the output signals. In panel (3) of Fig. 5, the presence of black dots signifies areas where the separation between the spectral components of the output signal does not match the FSR of the injected comb, indicating different

dynamical behavior for both modes, see Section 6 of the Supplementary Information. Although we did not perform such a systematic investigation experimentally, these features are well in line with our observations.”

In addition, within the revised caption of the new Fig. 5, we add detailed explanation about the black dots in panel (3):

“...while the black dots in panel (3) show where the FSR estimated from the output signal doesn't match the FSR of the injected comb, thus indicating a different type of dynamic.”

We believe that with these clarifications both in the caption and main text, the text ““FSR \neq 1, FSR=1” inside the panel (3) of the new figure becomes unnecessary. We have therefore removed it from the panel.

3) Other minor comments:

- a) p. 1, left, l. 16: "tunability. Meaning" -> "tunability, meaning"
- b) Fig. 2(d), the label of the axis: "Wavelength" -> "Wavelength"
- c) Fig. 5, l. 5 in the caption: "dashed line" -> "solid line" (?)
- d) p. 8, left, l. 4: "that is," -> "That is,"

We would like to thank the reviewer for pointing out these typos. We have applied the corresponding changes. Regarding the comment about the “dashed line” in Fig. 5 caption (line 5), we have kept it as it is, because here we are referring to the dashed lines in panels (1), (2), and (3) of Fig. 5 (a-d) in the new version of the figure. Note that the solid red lines are linked to panels (4) and (5) where we show the spectra of the injected and un-injected modes.